# Inhibitors of the Actin-Bundling Protein Fascin-1 Developed for Tumor Therapy Attenuate the T-Cell Stimulatory Properties of Dendritic Cells

**DOI:** 10.3390/cancers14112738

**Published:** 2022-05-31

**Authors:** Yanira Zeyn, Gregory Harms, Ingrid Tubbe, Evelyn Montermann, Nadine Röhrig, Maike Hartmann, Stephan Grabbe, Matthias Bros

**Affiliations:** 1Department of Dermatology, University Medical Center Mainz, Langenbeckstrasse 1, 55131 Mainz, Germany; yanira.zeyn@uni-mainz.de (Y.Z.); tubbe@uni-mainz.de (I.T.); monterma@uni-mainz.de (E.M.); n.roehrig@uni-mainz.de (N.R.); maike.hartmann@unimedizin-mainz.de (M.H.); stephan.grabbe@unimedizin-mainz.de (S.G.); 2Cell Biology Unit, University Medical Center Mainz, Langenbeckstrasse 1, 55131 Mainz, Germany; greharms@uni-mainz.de; 3Departments of Biology and Physics, Wilkes University, 84 W. South St., Wilkes Barre, PA 18766, USA

**Keywords:** Fascin-1, dendritic cells, CD86, CD273, NP-G2-044, BDP-13176

## Abstract

**Simple Summary:**

Expression of the actin-bundling protein Fascin-1 (Fscn1) is largely restricted to neuronal cells and to activated dendritic cells (DCs). DCs are important inducers of (antitumor) immune responses. In tumor cells, de novo expression of Fscn-1 correlates with their invasive and metastatic activities. Pharmacological Fscn1 inhibitors, which are currently under clinical trials for tumor therapy, were demonstrated to counteract tumor metastasis. Within this study, we were interested in better understanding the effects of Fscn1 inhibitors on DCs and discovered that two distinct Fascin-1 inhibitors affect the immune-phenotype and T-cell stimulatory activity of DCs. Our results suggest that systemic application of Fscn1 inhibitors for tumor therapy may also modulate antitumor immune responses.

**Abstract:**

Background: Stimulated dendritic cells (DCs), which constitute the most potent population of antigen-presenting cells (APCs), express the actin-bundling protein Fascin-1 (Fscn1). In tumor cells, de novo expression of Fscn1 correlates with their invasive and metastatic properties. Therefore, Fscn1 inhibitors have been developed to serve as antitumor agents. In this study, we were interested in better understanding the impact of Fscn1 inhibitors on DCs. Methods: In parallel settings, murine spleen cells and bone-marrow-derived DCs (BMDCs) were stimulated with lipopolysaccharide in the presence of Fscn1 inhibitors (NP-G2-044 and BDP-13176). An analysis of surface expression of costimulatory and coinhibitory receptors, as well as cytokine production, was performed by flow cytometry. Cytoskeletal alterations were assessed by confocal microscopy. The effects on the interactions of BMDCs with antigen-specific T cells were monitored by time lapse microscopy. The T-cell stimulatory and polarizing capacity of BMDCs were measured in proliferation assays and cytokine studies. Results: Administration of Fscn1 inhibitors diminished Fscn1 expression and the formation of dendritic processes by stimulated BMDCs and elevated CD273 (PD-L2) expression. Fscn1 inhibition attenuated the interaction of DCs with antigen-specific T cells and concomitant T-cell proliferation. Conclusions: Systemic administration of Fscn1 inhibitors for tumor therapy may also modulate DC-induced antitumor immune responses.

## 1. Introduction

Expression of the F-actin-bundling protein Fascin-1 (Fscn1) is tightly regulated at the transcriptional level [1,2] and, under homeostatic conditions, is largely confined to neuronal and glial cells, as well as some endothelial cell populations [3]. Fscn1 was demonstrated to be required for the growth and stabilization of axons in the case of neuronal cells [4] and to support the formation and turnover of filopodial extensions in other cell types [3]. Interestingly, de novo expression of Fscn1 was reported early for Epstein–Barr virus-infected B cells [5] and subsequently for human T-lymphotropic virus type 1-infected T cells [6]. In addition, immortalized tumor cell lines [7] were reported to express Fscn1 and have frequently been used to as cell models to delineate the interaction of Fscn1 with other cytoskeletal proteins, such as cofilin-1 [8,9] and Daam1 (disheveled-associated activator of morphogenesis 1) [10,11], among others [12].

Fscn1 may also be expressed de novo by tumor cells [13,14]. In response to the expression of Fscn1, tumor cells were reported to display increased migratory activity, thereby enhancing the invasive and metastatic properties of the tumor [15]. Accordingly, a number of studies have correlated poor prognosis to the extent of Fscn1 expression [16]. Besides its structural properties, Fscn1 was also shown to actively translocate from the cytoplasm into the nucleus of tumor cells and to positively regulate expression of pro-tumorigenic genes, such as the amino-acid transporter solute carrier family 3 member 2 [17] and to promote pro-tumorigenic canonical wingless signaling via activation of activation of focal adhesion kinase [18]. To date, several Fscn1 inhibitors that block interaction of Fscn1 with F-actin in order to inhibit tumor metastasis have been developed and successfully evaluated in vitro and in preclinical models [19,20,21]. More recently, a clinical phase I trial demonstrated that oral application of a Fscn1 inhibitor was well-tolerated and demonstrated antimetastatic activity, with increased progression-free survival in a number of patients [22]. Based on these results, a phase 2A clinical trial enrolling ovarian cancer patients to be treated with the Fscn1 inhibitor alone or in combination with a checkpoint inhibitor is planned.

Aside from constitutive expression of Fscn1 in some cell types and its de novo expression in virus-infected and malignant cells, we and others have previously shown that Fscn1 is strongly upregulated in stimulated dendritic cells (DCs) [23,24]. DCs are scattered throughout the body and constantly internalize extracellular material [25]. A fraction of DCs migrates into lymph nodes and the spleen to present derived oligopeptide antigens to T cells. Under homeostatic conditions, DCs induce peripheral T-cell tolerance towards self and harmless environmental antigens due to the absence of T-cell costimulatory signals. However, in response to pathogen-derived and endogenous danger signals, DCs are activated, upregulate surface expression of antigen-presenting receptors and costimulators and migrate to secondary lymphoid organs in elevated numbers to induce T-effector cells [26]. In that regard, activated DCs are the most potent antigen-presenting cell population and are the main inducers of primary immune responses directed, for example, against malignant cells [27].

We have previously shown that stimulation-induced Fscn1 in DCs is responsible for the formation of dendritic protrusions [23,24]. Dendritic protrusions may contribute to the migratory activity of activated DCs, as deduced from impaired migration of Hela cells in response to Fscn1 inhibition [7]. Subsequently, Yamakita and colleagues demonstrated that DCs of Fascin-1 knockout mice displayed attenuated migratory activity in vivo [28].

Previously, it was shown that inhibition of Fscn1 in stimulated DCs using antisense oligonucleotides attenuated their T-cell stimulatory capacity, although surface expression of major histocompatibility complex (MHC)II and costimulators remained unaltered [29]. We demonstrated that Fscn1 colocalized with F-actin and that this complex accumulated within the immunological synapse (IS) formed between DCs and antigen-specific T cells [30]. In the case of DCs, which lacked the costimulators CD80 and CD86, IS formation was weak and accompanied by strongly attenuated Fscn1/F-actin accumulation. Accordingly, Elizondo and colleagues reported that CD40-deficient DCs displayed attenuated Fscn1 expression in response to stimulation, which was associated with impaired DC/T-cell interactions and T-cell hypoproliferation [31]. Forced overexpression of Fscn1 rescued the attenuated T-cell stimulatory capacity of CD40-deficient DCs.

In light of the important role of DCs in the induction of (antitumor) T-cell responses and the well-established role of Fscn1 for the functional activity of activated DC in migration and T-cell activation, the intent of this investigation is to elucidate the impact of Fscn1 inhibitors developed for tumor therapy with DCs. Treatment of bone-marrow-derived DCs (BMDCs) with two structurally distinct Fscn1 inhibitors, NP-G2-044 and BDP-13176, interfered with the acquisition of a stimulation-induced mature immune phenotype, which was correlated with the applied dose (NP-G2-044 > BDP-13176). Interestingly, both inhibitors evoked PD-L2 expression in BMDCs. Of note, NP-G2-044 also attenuated antigen uptake by unstimulated BMDCs. Furthermore, both inhibitors diminished Fscn1 expression in stimulated BMDCs and attenuated both their interaction with antigen-specific CD4^+^ T cells and their T-cell stimulatory capacity, which was also apparent when applying Fscn1 inhibitors directly to DC/T-cell cocultures. However, both Fscn1 inhibitors also differentially affected T-cell activation by agonistic antibodies, suggesting unspecified off-target effects. Altogether, our results indicate that Fscn1 inhibitors developed for tumor therapy may also interfere with adaptive antitumor immune responses on several levels.

## 2. Materials and Methods

### 2.1. Materials

Fscn1 inhibitors NP-G2-044 (Selleckchem, Houston, TX, USA) and BDP-13176 (MedChemExpress, Monmouth Junction, NJ, USA) were reconstituted in DMSO (Roth, Karlsruhe, Germany). APC-eFl70-labeled anti-CD11c (clone N418), FITC-MHCI (28148), eFl450-MHCII (M5/114.15.2), APC-CD40 (1C10), PerCP-eFl710-CD80 (16-10A1), PE/Cy7-CD86 (GL-1), PE/TexasRed-CD274/PD-L1 (10F.9G2), PE-CD273/PD-L2 (Ty25), eFl506-CD3 (145-2C11), SB600-CD11b (M1/70), SB702-CD19 (eBio103), PE-NK1.1 (PK136), PE-eFl610-Ly6G (1A8-L6g), eFl780-FVD and eFl450-FVD used for flow cytometric analysis were purchased from BD Biosciences (Franklin Lakes, NJ, USA), BioLegend (San Diego, CA, USA) or ThermoFisher (Waltham, MA, USA). Unlabeled mouse anti-human Fscn1 antibody (clone 55K2; Sigma-Aldrich, Deisenhofen, Germany), a corresponding isotype control antibody (mouse IgG1, clone MOPC-21, BioLegend), secondary AF488-labeled IgG goat anti-mouse antibody (948492), AF647-labeled anti-CD11c antibody (clone N418), Hoechst (nuclear staining) and Alexa Fluor 555 phalloidin (F-actin) (all from ThermoFisher) were used for confocal laser scanning analysis (CLSM).

### 2.2. Mice

C57BL/6 mice, as well as OT-I [32] and OT-II [33] mice on C57BL/6 background, were bred and maintained in the Central Animal Facility of the Johannes Gutenberg-University Mainz under specific pathogen-free conditions on a standard diet according to the guidelines of the regional animal care committee. The “Principles of Laboratory Animal Care” (NIH publication no. 85-23, revised 1985) were followed. Mice (6–12 weeks) were sacrificed for organ retrieval according to § 4(3) TierSchG.

### 2.3. Cell Culture

Spleens were mechanically disrupted using a pestle and a 40 µM cell strainer (Greiner Bio-One, Frickenhausen, Germany) to obtain a single-cell suspension. Spleen cells (2 × 10^6^/500 µL) were cultured in FACS tubes overnight in medium (IMDM, 2 mM L-glutamine, 100 U/mL penicillin, 100 µg/mL streptomycin (all from Sigma-Aldrich, Deisenhofen, Germany) and 50 µM ß-mercaptoethanol (Roth) containing 5% FBS (PAN-Biotech, Aidenbach, Germany)). Bone marrow cells (2 × 10^5^/mL) were seeded in 12-well suspension culture plates (Greiner Bio-One) in culture medium supplemented with recombinant murine GM-CSF (10 ng/mL; Miltenyi, Bergisch Gladbach, Germany). Culture media were replenished on days 3 and 6 of culture.

### 2.4. Confocal Laser Scanning Microscopy (CLSM)

BMDCs (days 6–7 of culture) were incubated overnight with Fscn1 inhibitors, harvested and resuspended (10^6^/mL) in staining buffer. Cells were cytospun (4000 rpm, 5 min, room temperate) onto microscope slides (Superfrost Plus; VWR, Darmstadt, Germany) using a Cytospin 3 (Thermo Fisher, Waltham, MA, USA). Samples were incubated with pre-cooled methanol (Carl Roth) for 10 min for permeabilization of cell membranes and washed 2 times with PBS. Cytospins were incubated with PBS/2% bovine serum albumin (Sigma-Aldrich, Deisenhofen, Germany) plus 2.4-G2 antibody (1:50) for 10 min in a humified chamber at room temperature to block unspecified binding sites. Afterwards, samples were incubated with Fscn1-specific (diluted 1:50 in PBS/2% FBS) or isotype control (1:50 in PBS/2% FBS) for 20 min at room temperature in a humidified chamber. Samples were washed with PBS and incubated with secondary anti-mouse (1:400 in PBS/2% FBS), CD11c-specific antibody (1:50 in PBS/2% FBS) and phalloidin (1:150 in PBS/2%FBS) for 20 min at 4 °C in a humified chamber. After washing 2 times (PBS), samples were incubated with Hoechst dye (1 µg/mL) for 5 min and washed with purified water. In control settings, cytospins were left untreated or were incubated with one agent only. Finally, cytospins were covered with fluorescence mounting medium (DAKO; Agilent, Santa Clara, CA, USA).

Samples were imaged on a Leica SP8 confocal microscope (Mannheim, Germany) with a 20/0.75 NA air objective with exposure from a 405 nm laser for transmission images and for Hoechst excitation (emission and detection within a spectral window of 415 nm to 530 nm), with 488 nm laser exposure for AF 488 (Fscn1) excitation (emission and detection within a spectral window of 499 nm to 581 nm), 552 nm laser exposure for phalloidin plus 555 (F-Actin) excitation (emission and detection within a spectral window of 562 nm to 632 nm) and with 638 nm laser exposure for AF 647 (CD11c) excitation (emission and detection within a spectral window of 647 nm to 795 nm). The images were acquired at a factor of at least 2.3 times less than the calculated confocal resolution at a scan rate of 400 lines/min and with 2× averaging for 581 µm × 581 µm images, with a pixel size of 0.11 µm (or 5296 × 5296 pixels). All images used in comparison were prepared and acquired under the same conditions. Control images acquired of single stained cells or of unstained cells revealed that under these detection and imaging schemes and under the aforementioned conditions for imaging cross-talk fluorescence, background fluorescence and background autofluorescence were either non-existent or below statistical significance. The images shown in the figures were smoothed with the standard Leica smoothing algorithm. In some cases, the Hoechst signal was processed by lowering the upper threshold by as much as 20% in order to create homogenous nuclear images and homogenous cell nuclear borders for cell border determination. In some cases, the CD11c signal had up to a 10% cutoff and/or a 10% threshold applied in order to create a homogeneous cell border.

Once the cell borders were created, the original fluorescence images were then used for quantification. The fluorescence ratios were calculated from the total sum of each spectral fluorescence signal intensity for the defined regions of each cell. The nuclear and cell-bounded regions were obtained with automated batch processing analysis with Imaris software version 9.3.1 (Bitplane, Zurich, Switzerland) According to the cell biology package with the nuclear boundary obtained from the edge of the Hoechst image and with the cell boundary obtained from the edge of the CD11c image and the algorithm, there can be only one nucleus per detected cell. Cells without a visible nucleus or cell boundary were rejected from the analysis.

### 2.5. Surface Markers

Differentially treated spleen cells and BMDCs were washed in staining buffer (PBS, 1% FBS, 0.5 mM EDTA) and were incubated with rat anti-mouse CD16/CD32 antibody (clone 2.4G2; 15 min, room temperature) to block antibody binding to Fcγ receptors. Then, samples were incubated with fluorescence-labeled antibodies (20 min, 4 °C) and washed with PBS. Afterwards, samples were incubated with FVD to discriminate live/dead cells. Samples were measured using an Attune NxT flow cytometer (Thermo Fisher, Waltham, MA, USA), and data were analyzed using Attune NxT software (Thermo Fisher, Waltham, MA, USA).

### 2.6. Antigen Uptake and Processing

The capacity of BMDCs to internalize and process antigens was monitored by applying 25 µg/mL OVA-AF647 (uptake) and OVA-DQ (processing), respectively, to BMDCs differentiated in 12-well plates. Samples were incubated at 37 °C for 1 h. In parallel settings, replicate samples were preincubated on ice for 30 min prior to administration of OVA derivatives, followed by incubation on ice for 1 h as a control to differentiate temperature-insensitive binding and temperature-sensitive internalization. Then, samples were harvested, and CD11c-specific antibody was applied. Samples were subjected to flow cytometric analysis in order to delineate the frequencies of BMDCs internalization and OVA processing.

### 2.7. Cytokines

Supernatants of spleen cells and BMDCs differentially treated overnight with Fscn1 inhibitors, as well as those of DC/T-cell cocultures, were used to measure cytokine contents by flow cytometry (Cytometric Bead Array; BD, Heidelberg, Germany), followed by analysis using FCAP Array^TM^ software v.1 (BD, Heidelberg, Germany).

### 2.8. DC/T-Cell Interaction

BMDCs were incubated with 5 µg/mL of endotoxin-free ovalbumine (OVA; Merck, Darmstadt, Germany). After 3 h, Fscn1 inhibitors were applied at different concentrations as indicated. LPS (100 ng/mL; Merck Millipore, Burlington, MA, USA) was added 1 h later. The next day, samples were harvested and washed, and BMDCs (10^6^/mL) were resuspended in PBS. Splenic OVA-responsive CD4^+^ T cells (OT-II) were immunomagnetically sorted (Miltenyi Biotec, Bergisch-Gladbach, Germany) and resuspended in PBS (10^6^ T cells/mL). BMDC (CellTrace Violet) and T cells (carboxyfluorescein succinimidyl ester, CFSE) were labeled with the corresponding fluorescent dyes (5 µM each; both from Thermo Fisher, Waltham, MA, USA) for 30 min at 37 °C. After washing, BMDCs (2 × 10^5^/mL) and T cells (10^6^/mL) were resuspended in culture medium, and 150 µL of each cell suspension was applied to 8-well ibiTreat µ-slides (ibidi, Gräfeling, Germany).

BMDC/T-cell cultures were kept in an OkoLabs environmental incubator (H-301K environmental chamber, Oko Touch, Oko Pump, T-Control and CO_2_ control, Ottaviano, Italy) on a microscope table. DC/T-cell interaction was monitored by CLSM using a Leica TCS SP8 (Mannheim, Germany) equipped with a 20/0.75 NA objective (405 nm and 488 nm excitation, respectively, each approximately 150 µW; emission windows: 415–478 nm and 498–578 nm) and with scanning differential interference contrast transmission imaging in a 580 µm × 580 µm frame format (400 lines/s, 1.14 µm/pixel (512 × 512 pixels per frame)) and with two times averaging per line (frame acquisition of every 2 min per selected position within the chamber) over the 8 h observation period.

Image sequences were imported into Imaris version 9.3.1 (Bitplane, Zurich, Switzerland). DCs and T cells were detected automatically by fluorescence with whole-cell spot and whole-cell surface analysis (differed in control analysis by less than 1%). Whole-cell spot automated analysis was applied to all images (estimated cell diameter: 16 µm). Automated tracking was performed for all image sequences within Imaris using the autoregressive motion algorithm, with a maximum average distance of 60 µm to 300 µm per step and with zero step gap applied.

For T-cell/DC interaction analysis, a smoothed boundary around the DCs was created and expanded by the length of the average T-cell diameter. A T cell was considered to be in contact with a DC if the center of the T cell’s fluorescence was within the boundary of a DC. The average number of T cells per DC was determined for each DC and was tracked and reported as the average per image and per DC for the entire track length of each tracked DC. Cell velocity was determined by dividing each ‘x, y step’ by 2 min. Acceleration (or cell acceleration) was determined by subtracting a step speed from the previous step speed and dividing by 120 s (or 2 min). The displacement length was determined by subtracting the initial x, y position from the final x, y position to determine the difference in the vector length for each track over 8 h of acquisition. The track length added each absolute x, y vector step for an entire track over 8 h of acquisition. The mean track speed was acquired by averaging the speed of the steps for individual tracks. Results of cell tracking were verified using Fiji (plugin: TrackMate) [34].

### 2.9. T-Cell Proliferation

BMDCs were incubated with OVA, and 3 h later, Fscn1 inhibitors were applied at different concentrations as indicated. In parallel assays, LPS (1 µg/mL) was added 1h later. On the next day, samples were harvested, washed and resuspended in culture medium w/o GM-CSF. BMDCs were (2 × 10^4^/200 µL) seeded in wells (triplicates) of 96-well plates (Greiner Bio-One) and were serially titrated (1:2). Splenic OVA peptide-specific CD8^+^ (OT-I) and CD4^+^ T cells (OT-I) were immunomagnetically enriched as recommended by the manufacturer and were added (each 5 × 10^4^ cells/100 µL) to serially diluted BMDCs. In some experiments, T cells were polyclonally stimulated by applying beads conjugated with agonistic anti-CD3 and anti-CD28 antibodies (Dynabeads Mouse T-Activator CD3/CD28; Thermo Fisher, Waltham, MA, USA) as recommended by the manufacturer. After 3–4 days of culture, ^3^H-thymidine (0.5 μCi/well) was added to the cocultures for 16–18 h to assess T-cell proliferation. To this end, cell lysates were transferred onto glass fiber filter mats (Harvester 96; TomTec, Hamden, CT, USA). Genomically incorporated radioactivity was monitored using a microplate β-counter (1450 MicroBeta Trilux; Perkin Elmer, Waltham, MA, USA).

### 2.10. Statistical Analysis

Statistical analysis was performed using GraphPad Prism software v5.0 (GraphPad Software Inc., San Diego, CA, USA). Results were expressed as mean ± standard error of the mean (SEM). Differences among groups were tested by one-way ANOVA, followed by post hoc Tukey’s test, assuming significance at *p* < 0.05.

## 3. Results

### 3.1. NP-G2-044 Attenuates Expression of DC Activation Markers

First, we determined the effects of the two structurally distinct Fscn1 inhibitors, NP-G2-044 and BDP-13176, on primary splenic DC when applied at a range of concentrations previously reported to confer Fscn1 inhibition in tumor cells (10^−^^6^–5 × 10^−^^5^ M) [20]. At these concentrations, the viability of DCs, as well as of most of other splenic leukocyte populations, was not affected (Appendix A). Only B cells displayed strongly attenuated viability in response to treatment with the highest concentration of NP-G2-044 (5 × 10^−^^5^ M). Interestingly, whereas NP-G2-044 attenuated the expression of MHCII and the costimulatory receptors CD80 and CD86 by conventional (c)DC1/2 and plasmacytoid (p)DCs in a dose-dependent manner, BDP-13176 had no effect (Figure 1). Stimulation of spleen cells with the TLR4 agonist LPS plus the TLR7 agonist R848 to obtain maximal stimulation of either DC population yielded enhanced expression of both costimulatory receptors. In the case of coapplication of NP-G2-044, upregulation-enhanced expression was impaired at higher concentrations (10^−^^5^ M, 5 × 10^−^^5^ M). Cotreatment with BDP-13176 attenuated stimulation-induced upregulation of CD80 and CD86 only when applied at the highest concentration (5 × 10^−^^5^ M). Interestingly, neither the viability of other splenic leukocyte populations (Appendix A) nor basal or stimulation-induced expression of activation markers (not shown) was affected by either Fscn1 inhibitor, except for B cells, which were strongly affected by NP-G2-044 in both regards. In agreement with the significant inhibitory effect of NP-G2-044 on activation marker expression with splenic DC and B cells, it was observed that this agent also attenuated stimulation-induced cytokine production by spleen cells in a dose-dependent manner (Appendix A). BDP-13176 counteracted cytokine induction of stimulated spleen cells only at the highest concentration tested.

Due to the low abundance of primary DCs, we aimed to elucidate the effects of Fscn1 inhibitors in more detail with bone-marrow-derived DCs (BMDCs). The viability of DCs in response to treatment with Fscn1 inhibitors was only moderately affected upon application of NP-G2-044 at the highest concentration during DC stimulation (Figure 2A). Treatment of unstimulated DCs with NP-G2-044 and BDP-13176 had no major effect on basal expression of surface markers (Figure 2B). Most surface receptors were upregulated in response to stimulation with LPS. Interestingly, concomitant application of NP-G2-044 at lower concentrations moderately elevated expression of PD-L2, whereas higher concentrations interfered with LPS-induced upregulation of programmed death ligand (PD-L)1. Coapplication of LPS and BDP-13176 also tended to increase PD-L2 surface expression at most concentrations. Furthermore, LPS-conferred induction of proinflammatory cytokines was dose-dependently inhibited by NP-G2-044 in the cases of TNF-β, IL-6 and IL-10, and similar tendencies were observed for IL-1β (Figure 2C). With the exception of IL-1β, treatment with BDP-13176 did not inhibit cytokine production.

Antigen uptake is a hallmark of unstimulated DC and becomes largely inhibited during the course of activation [35]. Because both Fscn1 inhibitors interfered with the stimulation of DCs, we investigated potential effects of these agents on antigen uptake and processing. Pretreatment of unstimulated DCs with NP-G2-044 but not with BDP-13176 impaired temperature-dependent internalization of fluorescence-labeled ovalbumin (OVA), which was used as a model antigen (Figure 2D, left panel). Concomitant stimulation with LPS reduced OVA internalization for both groups to a similar extent. Processing of internalized OVA protein was somewhat diminished in the case of BMDC pretreatment with either Fscn1 inhibitor and further decreased to a larger extent in the case of concomitant stimulation (Figure 2D, right panel).

**Figure 2 cancers-14-02738-f002:**
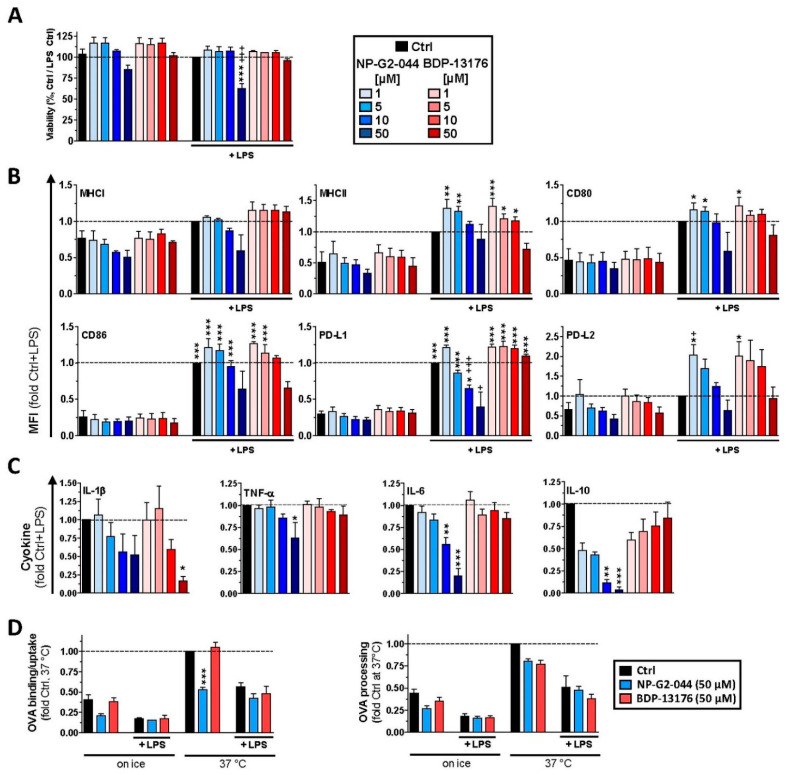
Fscn1 inhibitors interfere with stimulation-induced acquisition of an immune-stimulatory phenotype of bone-marrow-derived DCs (BMDCs). (**A**–**C**) BMDCs were incubated overnight with Fscn1 inhibitors as described (see Figure 1) and stimulated with LPS (100 ng/mL). (**A**,**B**) The next day, samples were subjected to flow cytometric analysis as described in [36]. Graphs denote the percentage of FVD-negative cells (**A**) or MFI (**B**) (mean ± SEM of three to four experiments) of marker expression relative to expression by the stimulated control (Ctrl + LPS). (**C**) Cytokine concentrations of BMDC culture supernatants were determined by CBA (mean ± SEM of 3–8 experiments). (**D**) Unstimulated BMDCs were incubated with Fscn1 inhibitors (each 5 × 10^−5^ M) as indicated, kept at a constant temperature as indicated, and OVA derivatives (OVA-AF647, OVA-DQ; each 25 µg/mL) were applied for the last 30 min of incubation. OVA uptake (left panel) and processing (right panel) were assessed by flow cytometry. Data denote the MFI (mean ± SEM of three experiments) and are presented relative to Ctrl. (**A**–**D**) Statistical differences are indicated: (**A**,**B**) vs. * Ctrl and ^+^ Ctrl + LPS, (**C**) vs. * Ctrl + LPS and (**D**) vs. * Ctrl (37 °C) (one-way ANOVA, Tukey test). *^,+^
*p* < 0.05, ** *p* < 0.01, ***^,+++^
*p* < 0.001.

### 3.2. Fscn1 Inhibitors Diminish Fscn1 Levels in Stimulated DCs but Affect F-Actin Contents Differentially

As expected, CD11c^+^ DCs in an unstimulated state expressed Fscn1 at rather low level, with strong upregulation in response to stimulation with LPS (Figure 3A,C, left panel). Furthermore, stimulated DCs also showed elevated levels of F-actin (Figure 3A,C, right panel). Stimulation of DCs in the presence of either Fscn1 inhibitor largely counteracted Fscn1 upregulation. However, whereas application of NP-G2-044 also strongly diminished F-actin contents, even below the level of unstimulated DCs, cotreatment with BDP-13176 yielded considerably enhanced F-actin levels. Interestingly, Fscn1 inhibitors prevented the formation of stimulation-induced dendritic processes (Figure 3B).

### 3.3. Fscn1 Inhibition Attenuates the Interaction of DCs with T Cells

Next, we investigated the consequences of Fscn1 inhibitor-dependent effects on Fscn1 expression and DC morphology with regard to their interaction with antigen-specific T cells. Both cell types were differentially labeled with live cell fluorescent dyes, and cell interactions were monitored by time-lapse confocal microscopy. Stimulated DCs were characterized by a higher velocity (Figure 4A) as compared to unstimulated DCs. Furthermore, DCs stimulated in the presence of NP-G2-044 displayed lower motility than stimulated DCs. Moreover, stimulated DCs contacted more T cells than unstimulated DCs (Figure 4B). On the contrary, DCs stimulated in the presence of either Fscn1 inhibitor contacted fewer T cells than LPS-treated DCs.

### 3.4. Fscn1 Inhibitors Impair the T-Cell Stimulatory Activity of DCs

Unstimulated DCs pretreated overnight with NP-G2-044 at higher concentrations induced lower proliferation of CD8^+^ (Appendix A) and CD4^+^ (Appendix A) T cells. Unstimulated and LPS-stimulated DCs pretreated with either Fscn1 inhibitor yielded no effect on cytokine levels in the case of CD8^+^ T cells. (Appendix A). In the case of CD4^+^ T-cell cocultures, only pretreatment of DCs with the highest dose of NP-G2-044 had a somewhat inhibitory effect on most cytokines monitored (Appendix A). LPS stimulation of DCs after inhibitor treatment resulted in attenuated CD4^+^ (Figure 4C) T-cell proliferation only at the highest dose of NP-G2-044. In most cases, cytokine levels were reduced when LPS-stimulated DCs cotreated with NP-G2-044 were used to activate T cells, often in a dose-dependent manner (Figure 4D). Moderately inhibitory effects were noted in the case of pretreatment with BDP-13176 for IL-9, IL-13 and IL-17, albeit below significance in the case of the former two cytokines.

Next, we addressed the efficacy of Fscn1 inhibitors to modulate the T-cell stimulatory capacity of DCs when applied during DC/T-cell coculture, as would also be the case in in vivo treatment. Under these conditions the extent of CD8^+^ and CD4^+^ T-cell proliferation evoked by unstimulated (Appendix A) and LPS-stimulated (Figure 5A,C) DCs decreased relative to the dose of either Fscn1 inhibitor, largely irrespective of the activation state of the DCs used as stimulators. The decrease in T-cell stimulation was accompanied by attenuated IFN-γ and TNF-γ contents of the corresponding CD8^+^ (Appendix A and Figure 5B) and CD4^+^ (Appendix A and Figure 5D) T-cell cocultures, reaching statistical significance only in the case of CD4^+^ T-cell-containing cocultures. In the case of NP-G2-044 application, the levels of these cytokines showed a tendency to inversely correlate with the applied dose, whereas in the presence of BDP-13176, attenuated cytokine levels were most obvious in the case of the highest Fscn1 inhibitor concentration (5 × 10^−^^5^ M). Interestingly, whereas NP-G2-044 at the lowest applied dose (10^−^^6^ M) exerted no effect on IL-9 and IL-13 produced by CD4^+^ T cells, production of both cytokines was almost abrogated above 10^−^^6^ M. On the contrary, application of BDP-13176 at lower concentrations yielded somewhat elevated levels of both cytokines, and concentrations of both cytokines were lower than in the Ctrl setting only at the highest concentration (5 × 10^−^^5^ M). DCs stimulated in the presence of NP-G02-044 and BDP-13176 at low concentrations tended to induce more IL-17 (below significance).

### 3.5. Fscn1 Inhibitors Coactivate CD8^+^ T Cells but Inhibit CD4^+^ T-Cell Activation

Due to the differential outcome of direct application of Fscn1 inhibitors to DC/T-cell cocultures (see Section 3.4) as compared to the use of corresponding pretreated DCs for coculture (see Section 3.3), especially in terms of cytokine production, we investigated the potential off-target effects of Fscn1 inhibitors on T cells. To this end, T cells were polyclonally stimulated in the presence of Fscn1 inhibitors. The metabolic activity of CD8^+^ T cells was strongly inhibited by NP-G2-044 when applied at the highest concentration (5 × 10^−^^5^ M) but moderately enhanced under treatment with BDP-13176 at the same dose (Figure 6A). Interestingly, both Fscn1 inhibitors tended to enhance CD8^+^ T-cell proliferation at lower doses, which was attenuated only in the presence of the highest dose of NP-G2-044 (Figure 6B). TNF-γ production was consistently abolished under the latter condition (Figure 6C). Interestingly, IFN-γ levels were higher with the application of NP-G2-044 below the highest concentration. A similar trend was observed in the case of treatment with BDP-13176. In agreement with the tendency of NP-G2-044 to promote CD8^+^ T-cell proliferation at low doses, we observed somewhat elevated expression of the early T-cell activation marker CD69 (Figure 6D, upper panel), whereas expression of the T cell activation marker CD25 was not affected under these conditions but reduced at higher NP-G2-044 concentrations and expression of CD62L as a marker of non-activated T cells remained largely unaltered (Figure 6D, lower panel).

Although the metabolic activity of polyclonally stimulated CD4^+^ T cells decreased with increasing doses of NP-G2-044, treatment with BDP-13176, except for at the highest dose, tended to yield moderately enhanced metabolic activity (Figure 6E). However, in general, CD4^+^ T-cell proliferation was inversely correlated with concentrations of either of the individual Fscn1 inhibitors (Figure 6F). Accordingly, NP-G2-044 treatment resulted in a dose-dependent decrease in levels of most cytokines (TNF-α, IL-9, IL-10, IL-13 and IL-17) (Figure 6G). However, IFN-γ levels were only somewhat enhanced, with the exception of the highest concentration of NP-G2-044. Although BDP-13176 had either no major effect on amounts of IFN-γ, IL-9 and IL-10 produced, TNF-γ concentrations were reduced at higher concentrations. Interestingly, BDP-13176 treatment showed a tendency to elevate IL-13 and IL-17 at the lowest treatment concentration, somewhat attenuating them at higher concentrations.

## 4. Discussion

The actin-bundling protein Fscn1 has attracted attention as a target protein in cancer therapy due to its requirement for tumor metastasis [15]. Accordingly, a number of pharmacological Fscn1 inhibitors have been developed, which impaired tumor cell migration in vitro by inhibiting the interaction of Fscn1 with F-actin [37,38] and accordingly attenuated tumor growth in preclinical mouse models [19]. To date, one clinical phase I trial that employed the Fscn1 inhibitor NP-G2-044 has been conducted and demonstrated efficacy in the treatment of various types of advanced and metastatic treatment-refractory tumors [22]. Consequently, a subsequent clinical trial proposal dedicated to determining the outcome of NP-G2-044 monotherapy and coapplied with a PD-(L)1-blocking antibody has been submitted (NCT05023486).

Besides de novo expression by malignant cells, Fscn1 expression is largely confined to neuronal cells, glial cells, some endothelial cells at low levels [3] and stimulated DCs, as shown by us and others [23,24]. Similar to tumor cells, DCs require Fscn1 to exert migratory activity [28] but also for their interaction with T cells [30] in order to mount antigen-specific T-cell responses [29]. In light of the importance of the patient immune system for the induction of antitumor responses, as well as other adaptive immune responses against pathogens, we investigated the potential effects of Fscn1 inhibitors on the DC immunophenotype and functions. We employed two distinct Fscn1 inhibitors, NP-G2-044 [37] and BDP-13176 [38]. Both Fscn1 inhibitors have been identified by screening of compound libraries for their activity to bind Fscn1 in order to block its F-actin cross-linking activity in a cell-free environment. However, only NP-G2-044 was further tested with regard to its inhibitory activity on tumor cell motility, and subsequently in tumor models [19,39]. We comparatively assessed the effects of these two structurally distinct inhibitors on DCs and T cells to delineate which effects were common and thereby most probably consequences of Fscn1 inhibition, as well as to which extent both inhibitors evoked distinct and thereby, most probably, Fscn1 inhibition-independent, inhibitor-specific, off-target effects.

We observed strong downregulation of Fscn1 in BMDCs cotreated with either Fscn1 inhibitor in the course of stimulation, which suggests that blockade of Fscn1 interaction with F-actin promoted its turnover. We are not aware of similar findings obtained concerning tumor cells treated with Fscn1 inhibitors. Somewhat surprisingly, however, in the case of treatment with NP-G2-044, diminished Fscn1 concentrations were accompanied by an overall decrease in F-actin, whereas in the case of BDP-13176 treatment, F-actin levels considerably increased. To the best of our knowledge, major effects of Fscn1 inhibition or knockdown on F-actin contents have not been described to date.

Fscn1 protein knockdown in stimulated DCs as conferred by either Fscn1 inhibitor was associated with the absence of the formation of dendritic protrusions, irrespective of F-actin contents. This observation is in agreement with the results of our previous studies, which demonstrated that Fscn1-specific inhibitory oligonucleotides inhibited the formation of dendritic processes [23,24]. Our observation of attenuated motility of stimulated DCs pretreated with either Fscn1 inhibitor in terms of velocity in cocultures with antigen-specific CD4^+^ T cells supports the role of Fscn1 in the migratory activity of DCs [28,39].

In these DC/T-cell coculture experiments, we also observed reduced interaction of CD4^+^ T cells with DCs stimulated in the presence of Fscn1 inhibitors. In line with these observations, such DCs exerted less proliferation of CD8^+^ and CD4^+^ T cells. We previously showed that Fscn1 accumulated within the IS between DCs and CD4^+^ T cells [30]. Akbar and colleagues reported on attenuated T-cell stimulatory activity of DCs pretreated with Fscn1 siRNA [29]. We and others have demonstrated that dynamic reorganization of F-actin within the IS on the DC side upon contact with a T cell is necessary for proper T-cell stimulation [30,40]. Therefore, our results confirm that functional impairment of Fscn1 interferes with DC-mediated T-cell activation. However, we also showed that both Fscn1 inhibitors inhibited the expression of MHCII and costimulators (CD80 and CD86) in bone-marrow-derived DCs used for most subsequent experiments and in primary splenic DC subpopulations (pDC and cDC1/2). Interestingly, at low to intermediate concentrations, both inhibitors promoted expression of PD-L2, which exerts coinhibitory effects by triggering PD-1 in T cells [41]. Therefore, we cannot rule out that, besides disturbed reorganization of F-actin, impaired expression of MHCII and costimulators may also contribute to the attenuated T cell stimulatory activity of corresponding pretreated DCs. Our observation of impaired expression of DC activation markers is in contrast to the results of a previous study showing that knockdown of Fscn1 in BMDCs by shRNA exerted no effect on the expression of MHCII and costimulators but was sufficient to attenuate T-cell stimulation [29]. Therefore, further studies are necessary to elucidate whether Fscn1 is implicated in expressional control of either gene or whether both inhibitors inhibit MHCII and costimulatory expression in DC via off-target effects. In agreement with the observation that Fscn1 inhibitors interfered with the acquisition of a mature phenotype of stimulated DC and their impaired interaction with antigen-specific T cells, they exerted less T-cell stimulatory activity.

Altogether, our results suggest that in vivo treatment with Fscn1 inhibitors for tumor therapy could affect the DC immunophenotype and DC/T-cell interactions. This issue has been addressed most recently in a study by Wang and colleagues [39]. Treatment of tumor-burdened mice with NP-G2-044 increased cDC1 and cDC2 frequencies within the tumor microenvironment (TME). As delineated for cDC1, this effect was strongly elevated upon cotreatment with a PD-1-blocking antibody. Under the latter condition, total DCs retrieved from the TME also displayed stronger expression of costimulatory receptors (CD40, CD80 and CD86) as compared to tumors derived from untreated mice and in response to a single treatment. Furthermore, Wang and colleagues demonstrated an inhibitory effect of NP-G2-044 on the migratory capacity of cDC1-like cells in vitro (MuTu cell line [42]. Therefore, the authors concluded that Fscn1 inhibition resulted in a migratory arrest of mature DCs in the TME. 

Cotreatment of tumor-burdened mice with NP-G2-044 and a PD-1-blocking antibody resulted in slower tumor growth and longer overall survival [39]. The corresponding TME contained higher frequencies of CD4^+^ and CD8^+^ T cells, and a higher frequency of the latter displayed an effector phenotype as compared to mice that received a single treatment with either antitumor agent.

Based on these results the authors hypothesized that mature DCs trapped within the TME as consequence of NP-G2-044 treatment may stimulate tumor antigen-specific T cells in local tertiary lymphoid structures (TLS), resulting in antitumor T-cell responses.

Some of our results are in contrast to these conclusions. First, we consider it conceivable that systemic application of the Fscn1 inhibitor (applied by gavage) could inhibit the motility of stimulated DCs throughout the body. In light of the limited life span of stimulated DCs [43], it is therefore conclusive that effective systemic Fscn1 inhibition would counteract DC accumulation within the TME. Secondly, T-cell activation requires presentation of internalized and processed antigens. Wang and colleagues demonstrated that DC-like MuTu cells treated with NP-G2-044 (3–10 µM) displayed an elevated increase in the uptake of the model antigen bovine serum albumin, as well as of dextran [39]. In contrast, we observed an inhibitory effect of this Fscn1 inhibitor (but not of BDP-13176) on the internalization of OVA by DC. However, due to the very low expression level of Fscn1 by unstimulated DCs, we cannot rule out that the results obtained for unstimulated DCs treated with NP-G2-044 is the consequence of an Fscn1-independent off-target effect. Likewise, it is not clear at to which extent unstimulated MuTu cells express Fscn1. Thirdly, our findings of attenuated interaction of Fscn1 inhibitor-pretreated DCs with T cells and the impaired T-cell stimulatory capacity of the former, both in the case of DC pretreatment, as well as when applied to DC/T-cell cocultures, do not support the hypothesis of enhanced T-cell stimulation by mature DCs trapped in the TME. Here, we also observed that levels of Tc1/Th1-associated cytokines (IFN-γ, TNF-γ), which are beneficial for antitumor immune responses [44], were generally attenuated in a largely Fscn1 inhibitor dose-dependent manner in corresponding DC/T-cell cocultures.

Altogether, these issues raise the question of whether NP-G2-044 may confer antitumor activity in vivo by alternative mechanisms. In this regard, it is noteworthy that within the TME, besides DCs tumor-associated macrophages (TAMs) also expressed Fscn1 at a high level [39]. In light of the important role of TAMs in conferring tumor progression, e.g., by neoangiogenesis, and in the release of anti-inflammatory mediators, such as IL-10, to promote tumor immune evasion [45], it is possible that inhibition of Fscn1 in TAMs may contribute to inhibition of tumor growth. Additional studies should elucidate the functional role of Fscn1 for TAMs and the consequences of its inhibition for these immunoregulatory cells.

Furthermore, the finding of elevated numbers of activated T cells within the TME of NP-G2-044-treated mice [39] may also be explained by our observation that NP-G2-044 but not BDP-13176 acted as a coactivator of polyclonally stimulated CD8^+^ T cells with regard to IFN-γ production. This result shows that NP-G2-044 may exert pronounced off-target effects on Fscn1-deficient (immune) cells via yet unknown molecular mechanisms.

Along this line, Wang and colleagues reported that besides DCs, other types of innate immune cells (neutrophils, monocytes and NK cells), which do not express Fscn1, were significantly enriched in the TME of NP-G2-044 treated mice. It remains possible that these cell populations may contribute to the observed pronounced antitumor effects.

As outlined above, the finding of tumoricidal activity of NP-G2-044, especially in combination with blockade of the PD-1/PD-L1 axis, could also be due to off-target effects of this Fscn1 inhibitor on non-DCs. It remains possible that each Fscn1 inhibitor may evoke distinct off-target effects, as evidenced by the inhibitory versus stimulatory effect of NP-G2-044 and BDP-13176 on F-actin levels, respectively, as well as the coactivating potential of NP-G2-044 in CD8^+^ T cells. Furthermore, we also noted that treatment of DC/T-cell cocultures with BDP-13176 but not NP-G2-044 at low to intermediate concentrations favored production Th2- (IL-13) and Th9- (IL-9) [46] associated cytokines, whereas NP-G2-044 at the lowest concentration upregulated Th17-associated IL-17 [47] production.

Altogether, our results confirm the necessity of performing comparative, in-depth in vivo studies employing distinct Fscn1 inhibitors and subsequent ex vivo analysis to delineate, in detail, by which mechanisms these may inhibit tumor growth. These mechanisms comprise inhibition of Fscn1-dependent tumor cell migration [19,20,21,48] and Fscn1-dependent gene expression in tumor cells [18,49,50,51], which may impact the characteristics of the TME, e.g., via soluble mediators. However, as shown in this study, Fscn1 inhibitors may also affect the immunophenotype and function of DCs and exert pronounced off-target effects on immune cells in an inhibitor-specific manner, as shown here for DCs and T cells.

## 5. Conclusions

In this article, we focused on the effects of pharmacological Fscn-1 inhibitors (NP-G2-044 and BDP-13176) on DCs. We demonstrated that application of either inhibitor prior to stimulation decreased Fascin-1 expression of activated BMDCs, as well as the formation of dendritic protrusions. Inhibition of Fscn1 attenuated T-cell/DC interactions and reduced the T-cell stimulatory activity of DCs. Furthermore, immunophenotypic analyses revealed enhanced expression of PD-L2 and diminished expression of costimulatory receptors and proinflammatory cytokines.

## Figures and Tables

**Figure 1 cancers-14-02738-f001:**
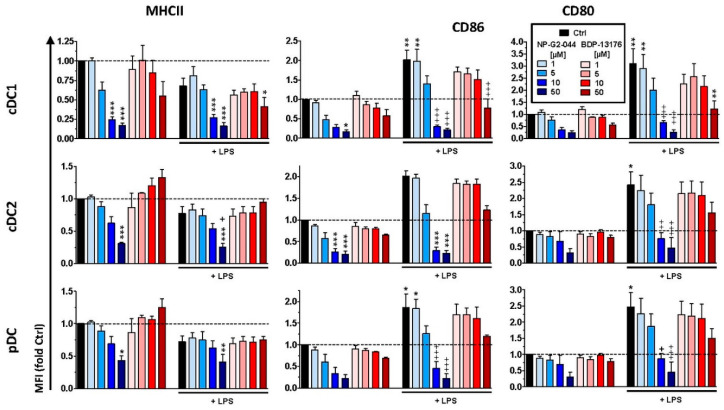
Fscn1 inhibitor NP-G2-044 impairs the activation state of splenic DC populations. Spleen cells were incubated with Fscn1 inhibitors (NP-G2-044 and BDP-13176) at different concentrations as indicated or DMSO as solvent control (Ctrl). In parallel settings, spleen cells were stimulated (LPS: 100 ng/mL, imiquimod: 1 µg/mL) 45 min after the corresponding Fscn1 inhibitor. The next day, expression of MHCII, CD80 and CD86 by different DC populations (cDC1, cDC2 and pDC) was assessed by flow cytometric analysis. The gating employed strategy is depicted in Appendix A. Graphs denote the fluorescence intensities (MFI) (mean ± SEM of four experiments) of marker expression. Statistical differences vs. * Ctrl and ^+^ Ctrl+LPS are indicated (one-way ANOVA, Tukey test). *^,+^
*p* < 0.05, **^,++^
*p* < 0.01, ***^,+++^
*p* < 0.001.

**Figure 3 cancers-14-02738-f003:**
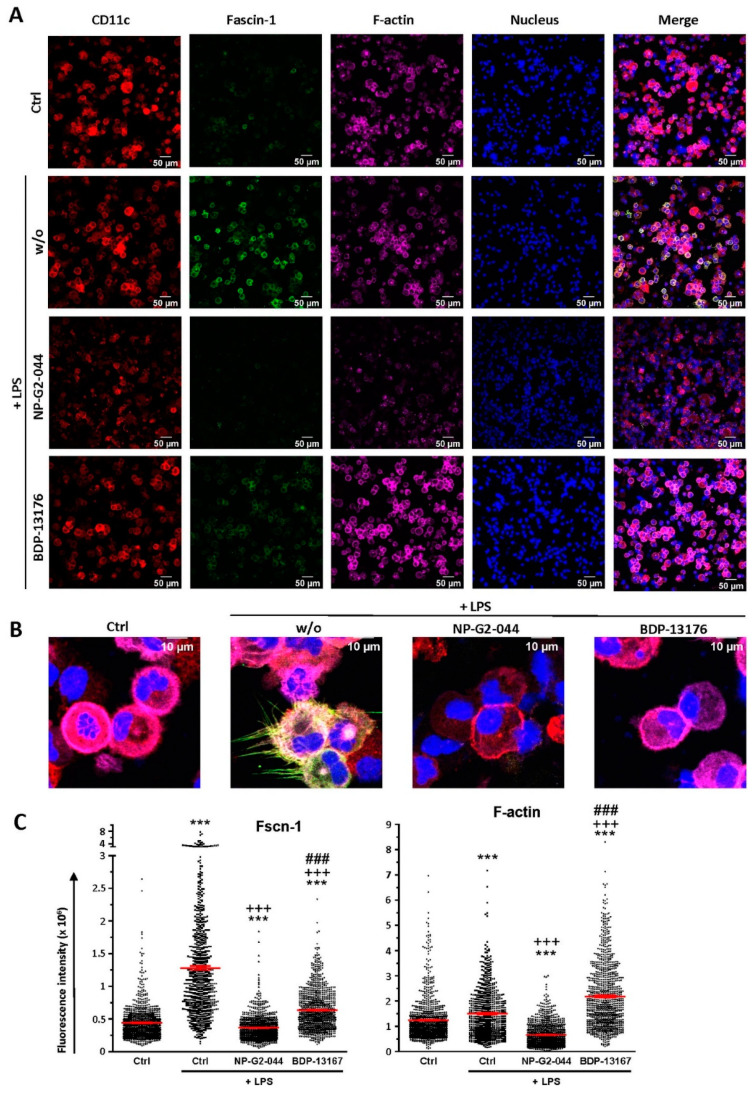
Fscn1 inhibitors diminish Fscn1 expression in stimulated DCs. DC were incubated with Fscn1 inhibitors (NP-G2-044, BDP-13176; each 5 × 10^−^^5^ M), followed by treatment with LPS (100 ng/mL) as indicated. Cytospins of differentially treated DCs were incubated with Hoechst dye for nuclear staining, fluorescence-labeled phalloidin for F-actin detection and specific antibodies for CD11c and Fscn1. An overview (**A**) and single cells (**B**) are shown. Images are representative of four independent experiments. (**C**) Quantification of Fscn1 (left panel) and F-actin (right panel) contents in differentially treated DCs. Data denote the corresponding fluorescence intensities per cell and the mean ± SEM of 789–1077 cells per group compiled from four independent experiments. (**C**) Statistical differences versus * Ctrl, versus ^+^ Ctrl + LPS and versus ^#^ NP-G2-044 + LPS are indicated (one-way ANOVA, Tukey test). ***^,+++,###^
*p* < 0.001.

**Figure 4 cancers-14-02738-f004:**
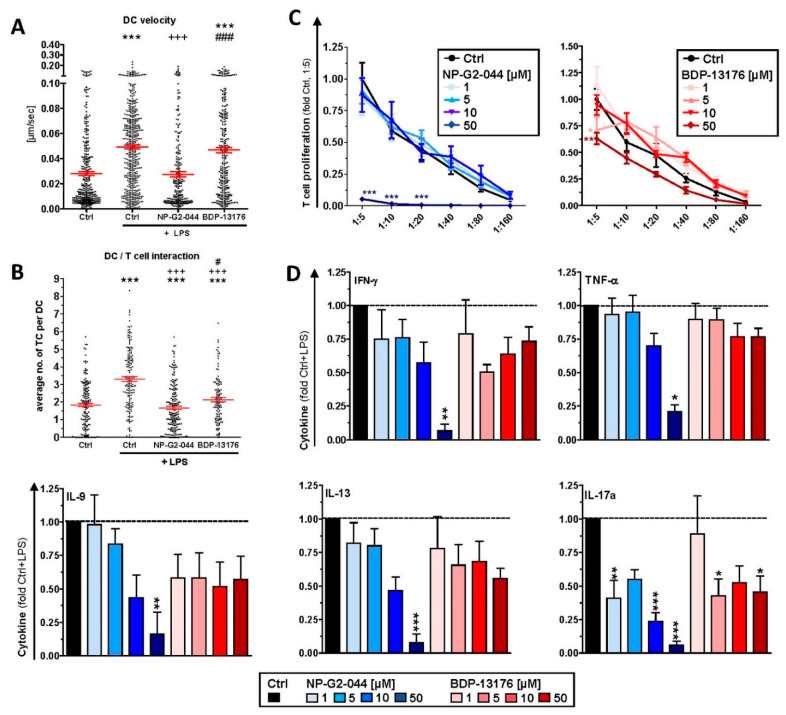
DCs pretreated with Fscn1 inhibitors display attenuated motility and T-cell interaction. DCs were incubated with OVA protein. After 3 h, Fscn1 inhibitors (NP-G2-044 and BDP-13176) and DMSO (Ctrl) were added as indicated. In parallel settings, LPS (100 ng/mL) was applied 45 min later as indicated. (**A**,**B**) The next day, DC^−^ (CellTrace Violet) and OVA-responsive CD4^+^ T cells (CFSE) were labeled with fluorescent living cell markers and cocultured (3 × 10^4^ DC/1.5 × 10^5^ T cells). (**A**) Motility of DC was tracked over an observation period of 8 h. The graph shows the velocity of single DCs and denotes the according mean ± SEM of 247–561 cells per group compiled from four experiments. (**B**) Quantification of DC/T-cell interactions. The graph displays the average number of DC-contacting T cells expressed as mean ± SEM of 125–260 cells per group compiled from four independent experiments. (**C**,**D**) The next day, serially titrated numbers of harvested and washed DCs (starting with 10^4^/100 µL) were cocultured with immune magnetically sorted OVA peptide-specific CD4^+^ (OT-II) T cells (5 × 10^4^/100 µL) in triplicate in 96-well plates. Proliferation of (**C**) CD4^+^ T cells was assessed by incorporation of ^3^H-thymidin applied for the last 16 h of 3–4 days of coculture. Data denote the mean ± SEM of three to five compiled experiments performed in triplicate relative to the corresponding Ctrl condition (1:5). (**D**) Prior to application of ^3^H-thymidine aliquots, DC/T-cell coculture supernatants (1:5) were retrieved, and cytokine contents in CD4^+^-containing cocultures were assayed by CBA. Data denote the mean + SEM of five experiments, with values normalized to Ctrl + LPS in each experiment. (**A**–**D**) Statistical differences versus * Ctrl, (**A**,**B**) vs. ^+^ Ctrl + LPS and vs. ^#^ NP-G2-044+LPS are indicated (one way ANOVA, Tukey test). * *p* < 0.05, ** *p* < 0.01, ***^,+++,###^
*p* < 0.001.

**Figure 5 cancers-14-02738-f005:**
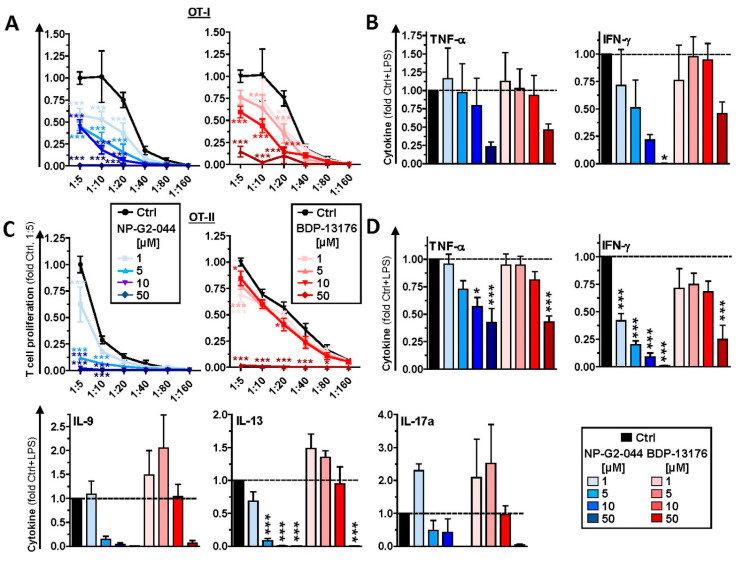
Results of application of Fscn1 inhibitors during DC/T-cell coculture in impaired T-cell proliferation. DCs were incubated with OVA protein (5 µg/mL). In parallel settings, LPS (100 ng/mL) was applied 45 min later. The next day, samples were harvested and washed, and serially titrated numbers of DCs were cocultured with OVA peptide-specific CD8^+^ (OT-I) (**A**,**B**) and CD4^+^ (OT-II) (**C**,**D**) T cells as described (see Figure 4). Fscn1 inhibitors were applied to DC/T-cell cocultures as indicated. Proliferation of (**A**) CD8^+^ and (**C**) CD4^+^ T cells was assessed. Data denote the mean ± SEM of two compiled experiments performed in triplicate relative to the corresponding Ctrl condition (1:5). Cytokine contents in (**B**) CD8^+^- and (**D**) CD4^+^-containing cocultures were assayed by CBA. Data denote the mean ± SEM of three experiments relative to Ctrl + LPS. (**A**–**D**) Statistical differences versus * Ctrl are indicated (one-way ANOVA, Tukey test). * *p* < 0.05, ** *p* < 0.01, *** *p* < 0.001.

**Figure 6 cancers-14-02738-f006:**
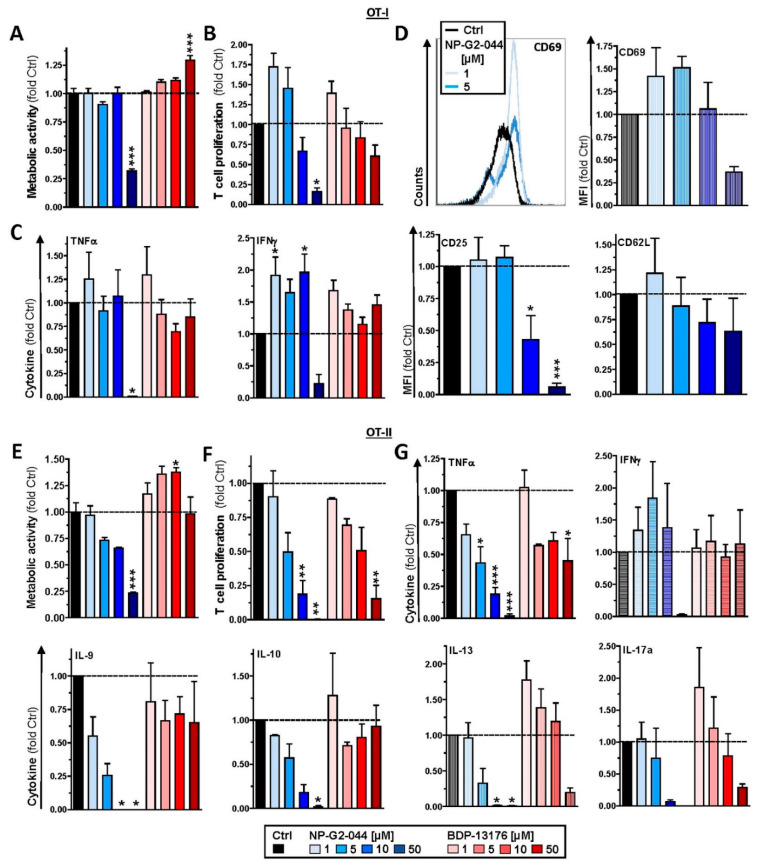
Fscn1 inhibitors promote CD8^+^ T-cell activation but attenuate CD4^+^ T-cell activation. Immunomagnetically sorted (**A**–**D**) CD8^+^ and (**E**–**G**) CD4^+^ T cells (each 5 × 10^4^/100 µL) were polyclonally stimulated using agonistic anti-CD3 (1 µg/mL) plus anti-CD28 (2 µg/mL) antibodies in the presence of Fscn1 inhibitors and DMSO (Ctrl) as indicated (**A**–**G**). (**A**,**E**) After 3–4 days of culture, metabolic activity was detected by application of MTT assay reagent. (**B**,**F**) T-cell proliferation and (**C**,**G**) cytokine concentrations were assayed as described (see Figure 4). (**D**) Left panel: expression of CD25, CD62L and CD69 was assessed by flow cytometric analysis. The graph is representative of six experiments. Right panel: quantification of T-cell proliferation. (**A**–**G**) Data denote the mean ± SEM of three to six compiled experiments relative to Ctrl. Statistical differences versus * Ctrl are indicated (one-way ANOVA, Tukey test). * *p* < 0.05, ** *p* < 0.01, *** *p* < 0.001.

## Data Availability

Not applicable.

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
