# Peer review of "Inhibitors of the Actin-Bundling Protein Fascin-1 Developed for Tumor Therapy Attenuate the T-Cell Stimulatory Properties of Dendritic Cells"

_cancers, 2022, doi:10.3390/cancers14112738_

Round 1
Reviewer 1 Report
Comments
In the present study, authors explored the potential effects of FSCN1 inhibitors NP-G2 and BDP on DCs and the DC-induced anti-tumor immune responses. Their findings showed that FSCN1 inhibitors reduced FSCN1 expression, inhibited dendritic processes, elevated PD-L2, and attenuated T cell proliferation. Accordingly, they suggest that systemic administration of Fscn1 inhibitors may influence DC-induced anti-tumor immune responses. Generally, this study has interests, experiments have been properly conducted, and most conclusions can be supported by the results. Hence, several suggestions are provided to improve this study.
1. The effects of two FSCN1 inhibitors on biological activities of DC are clearly different. Authors may briefly interpret their individual molecular acts.
2. In Figure 5B, comparing to control, are the changes of TNFa and IFNg production in response to FSCN1 inhibitors treatment significant?
3. Since authors concern that application of FSCN1 inhibitors may affect the DC-induced T cell-mediated anti-tumor activity, in vivo study could provide further and clear evidences to clarify the concern.
Author Response
- The effects of two FSCN1 inhibitors on biological activities of DC are clearly different. Authors may briefly interpret their individual molecular acts.
We have added a text passage in the 2nd paragraph of the discussion section to clarify that both Fscn1 inhibitors have been identified in the course of library screenings, based on their ability to block Fscn1-mediated F-actin bundling. To the best of our knowledge no further studies concerning additional interaction of these inhibitors with other proteins that could explain off target effects have been performed.
- In Figure 5B, comparing to control, are the changes of TNFa and IFNg production in response to FSCN1 inhibitors treatment significant?
The change in IFN-g levels at the highest concentration of NP-G2-044 was statistical significant. We have corrected this mistake.
- Since authors concern that application of FSCN1 inhibitors may affect the DC-induced T cell-mediated anti-tumor activity, in vivo study could provide further and clear evidences to clarify the concern.
We fully agree with the reviewer and have added an according paragraph at the end of the discussion section. Actually, we plan to perform according experiments. However, due to the duration of the approval process for animal experiments (>6 months from application to approval) and the necessity to provide data of in vitro experiments to substantiate the rationale for the planned in vivo studies, we aim to publish the results obtained so far.
Reviewer 2 Report
The authors comprehensively evaluated the impacts of two Fscn1 inhibitors (NP-G2-044, BDP-27 13176) on activities of DC as well as T cells . They showed that administration of Fscn1 inhibitors diminished Fscn1 expression and the formation of dendritic processes by stimulated BMDC, and elevated CD273 (PD-L2) expression. Moreover, Fscn1 inhibition attenuated the interaction of DC with antigen-specific T cells, and concomitantly T cell proliferation. Though the expreimental data is convincing, the ways the authors present data, make conclusion and discussion need to be improved before publication.
- What conclustions from current in vitro data agree or disagree with the in vivo study of Wang and coworkers in Ref. [39]? Almost two full pages in discussion are spent for describing the study of Wang and coworkers i.e. ref. 39. However, the words are redundant and that question is not discussed explicitly. This part should be sharpen. The author should precisely and breifly discuss the agreement and disagreement between these two studies. Moreover, the author should highlight what different insights they provide.
- To my understanding, Ref [39] performed in vivo studies on the same context, and the current conclusions are made based purely on in vitor studies. Two questions should be answered, what is new we can learn from the in vitro data? Should in vivo studies be conducted to support the new conclusions?
- The Figures and restuls are designed to show and compare the impacts of two inhibitors together. However, nearly no mechanism is explained for the differences. Moreover, this comparision contributes little into the current conclusions, and builds confusions. This should be improved.
- it is better to explicitly describe conclusions and discussion. For example, in the title, 'affect' is not informative. what kind of effects? suppress or enhance? The author should avoid this type of conculsion.
- Manuscript should be checked throughly for typos. Line 322: 'pProgrammed death-ligand (PD-L)1.'; Line 412: 'yielded n effect'; Line 420: in case o the former.
Author Response
- What conclustions from current in vitro data agree or disagree with the in vivo study of Wang and coworkers in Ref. [39]? Almost two full pages in discussion are spent for describing the study of Wang and coworkers i.e. ref. 39. However, the words are redundant and that question is not discussed explicitly. This part should be sharpen. The author should precisely and breifly discuss the agreement and disagreement between these two studies. Moreover, the author should highlight what different insights they provide.
We have restructured and condensed the discussion to present in a concise manner the differences between our results and those of the study of Wang and coworkers, and aim to interpret these results.
- To my understanding, Ref [39] performed in vivo studies on the same context, and the current conclusions are made based purely on in vitor studies. Two questions should be answered, what is new we can learn from the in vitro data? Should in vivo studies be conducted to support the new conclusions?
We also aim to perform in vivo studies to address this issue, and have added a paragraph at the end of the discussion to underscore the necessity to perform according experiments. We opted to present our in vitro findings in this study also due to the fact that the duration of the approval process for animal experiments is quite long (>6 months from application to approval) and that we have to provide supportive (published) results of in vitro experiments to substantiate the rationale for the planned in vivo studies. We also present more clearly the main findings of our in vitro results.
- The Figures and restuls are designed to show and compare the impacts of two inhibitors together. However, nearly no mechanism is explained for the differences. Moreover, this comparision contributes little into the current conclusions, and builds confusions. This should be improved.
We have added a statement in the 2nd paragraph of the discussion section to indicate that the Fscn1 inhibitors used in our study have been identified by library screening, based on their ability to block Fscn1-mediated F-actin bundling, but have not been tested any further with regard to potential engagement of additional proteins. We state that we therefore tested both compounds in parallel to delineate common (Fscn1 inhibition-dependent) and inhibitor-specific (off target) effects.
- it is better to explicitly describe conclusions and discussion. For example, in the title, 'affect' is not informative. what kind of effects? suppress or enhance? The author should avoid this type of conculsion.
We have altered the text to express more clearly the character of the observations in the title, headings and figure legends.
- Manuscript should be checked throughly for typos. Line 322: 'pProgrammed death-ligand (PD-L)1.'; Line 412: 'yielded n effect'; Line 420: in case o the former.
We have thoroughly corrected the manuscript.